# Dynamic rotation of the protruding domain enhances the infectivity of norovirus

**Chihong Song**[1☯], **Reiko Takai-Todaka**[2☯], **Motohiro Miki**[3], **Kei Haga**[2], **Akira Fujimoto**[2], **Ryoka Ishiyama**[2], **Kazuki Oikawa**[2], **Masaru Yokoyama**[4], **Naoyuki Miyazaki**[5,6], **Kenji Iwasaki**[5,6], **Kosuke Murakami**[4], **Kazuhiko Katayama**[2,4]*, **Kazuyoshi Murata**[1]*

**1** National Institute for Physiological Sciences, Okazaki, Japan, **2** Laboratory of Viral Infection I, Department of Infection Control and Immunology, Ōmura Satoshi Memorial Institute & Graduate School of Infection Control Sciences, Kitasato University, Tokyo, Japan, **3** DENKA Company Limited, Tokyo, Japan, **4** National Institute of Infectious Diseases, Tokyo, Japan, **5** Institute for Protein Research, Osaka University, Suita, Japan, **6** Life Science Center for Survival Dynamics, Tsukuba Advanced Research Alliance, University of Tsukuba, Tsukuba, Japan

☯ These authors contributed equally to this work.
* katayama@lisci.kitasato-u.ac.jp (KK); kazum@nips.ac.jp (KM)

**Data Availability Statement:** All relevant data are within the manuscript and its Supporting Information files. Cryo-EM maps have been deposited in the Electron Microscopy Data Bank

## Abstract

Norovirus is the major cause of epidemic nonbacterial gastroenteritis worldwide. Lack of structural information on infection and replication mechanisms hampers the development of effective vaccines and remedies. Here, using cryo-electron microscopy, we show that the capsid structure of murine noroviruses changes in response to aqueous conditions. By twisting the flexible hinge connecting two domains, the protruding (P) domain reversibly rises off the shell (S) domain in solutions of higher pH, but rests on the S domain in solutions of lower pH. Metal ions help to stabilize the resting conformation in this process. Furthermore, in the resting conformation, the cellular receptor CD300lf is readily accessible, and thus infection efficiency is significantly enhanced. Two similar P domain conformations were also found simultaneously in the human norovirus GII.3 capsid, although the mechanism of the conformational change is not yet clear. These results provide new insights into the mechanisms of non-enveloped norovirus transmission that invades host cells, replicates, and sometimes escapes the hosts immune system, through dramatic environmental changes in the gastrointestinal tract.

## Author summary

The capsid structure of caliciviruses has been reported to be classified into two different types, according to the species and genotype. One is the rising type of P domain conformation as shown in human norovirus GII.10 and rabbit hemorrhagic disease virus (RHDV), where the P domain rises from the S domain surface. The other is the resting type of P domain conformation as shown in human norovirus GI.1, sapovirus and San Miguel sea lion virus (SMSV), where the P domain rests upon the S domain. Here, we demonstrate that the P domain of the murine noroviruses changes reversibly between the rising and resting P domain conformation types in response to aqueous conditions. We

under accession numbers EMD-9735, EMD-9736, EMD-9737, EMD-9738, EMD-9739, EMD-9740 and EMD-9741. The atomic models have been deposited in the Protein Data Bank under accession number 6IUK.

**Funding:** This work was supported by AMED (Grant No. 18fk0108051h to KK; JP18fk0108034h and JP18am0101072j to KMurat), MEXT KAKENHI (Grant No. JP26102545 and JP16H00786 to KMurat), JSPS KAKENHI (to KK), and the collaborative programs for National Institute for Physiological Sciences (to KK). This research was partially supported by Platform Project for Supporting Drug Discovery and Life Science Research (Basis for Supporting Innovative Drug Discovery and Life Science Research (BINDS)) from AMED under Grant No. JP18am0101001 (support number 1161 to KMurat). The funders had no role in study design, data collection and analysis, decision to publish, or preparation of the manuscript.

**Competing interests:** The authors have declared that no competing interests exist.

also found the similar two P domain conformations in human norovirus GII.3 VLPs at the same time. Our findings provide new insights into the mechanisms of viral infection of caliciviruses.

## Introduction

Human norovirus (HNoV) is a major cause of epidemic nonbacterial gastroenteritis worldwide [1]. HNoV often causes severe sporadic infections, especially in nurseries and nursing homes. However, there are no efficient treatments or vaccines due to lack of a robust cultivation system and model animals for studying this virus [2]. Murine norovirus (MNoV), a species of norovirus affecting mice, was identified in 2003 [3]. MNoV is a unique norovirus which propagates in cell lines [3, 4], and shares genetic features with HNoV and has biochemically similar properties [5]. For these reasons, MNoV has been widely used as a model for HNoV infection.

Norovirus (NoV) is a non-enveloped, positive-stranded RNA virus belonging to the *Caliciviridae* family [6]. The common capsid structure of 180 copies of the VP1 capsid protein forming a T = 3 icosahedral particle is shared by all identified caliciviruses, which was first reported with HNoV GI.1 virus-like particle (VLP) produced by a baculovirus expression system in 1999 [7]. To form the capsid structure, VP1 is placed in three quasi-equivalent positions of asymmetric units designated as A, B, and C monomers. The VP1's A and B dimers (A/B dimers) are located around the icosahedral fivefold axes, while the VP1's C and C dimers (C/C dimers) are placed on the icosahedral twofold axes [8]. The VP1 protein consists of a shell (S) and protruding (P) domain. The S domain, consisting of an eight-stranded β sandwich structure, forms a contiguous icosahedral backbone to protect the central viral genome and has high amino acid sequence homology within the *Caliciviridae* family [9]. On the other hand, the P domain extended from the S domain via a flexible hinge loop further divided into a lower P1 subdomain and an upper P2 subdomain. With respect to the amino acid sequence, the P1 subdomain can be divided into two parts, the N-terminal P1 part (P1-1) and the C-terminal P1 part (P1-2), with the P2 subdomain interposed. The P2 subdomain is exposed on the top of the P domain and functions for virus attachment to cells [10].

The P domain in caliciviruses shows two conformations [11]. The first, called here the rising conformation, is shown in MNoV-1 [12], HNoV GII.10 [13], and rabbit hemorrhagic disease virus (RHDV) [14, 15], where the P domain rises from the S domain surface, forming an outer second shell. The second, called here the resting conformation, is shown in HNoV GI.1 [7], sapovirus [9], San Miguel sea lion virus (SMSV) [16] and feline calicivirus (FCV) [17, 18], where the P domain rests upon the S domain, forming an integrated shell with the S domain. The potential for dynamic structural changes in capsids has been discussed for viral infection and replication, but no direct evidence has been found thus far [11]. During the preparation of this manuscript, Snowden and colleagues also reported the resting conformation of MNoV-1 infectious particles in several samples, including heat treatment, suggesting a potential for the conformational changes in a single virus [19], although the mechanism has not yet been elucidated.

Here, using cryo-electron microscopy (cryo-EM), we show that the P domain of MNoV infectious particles reversibly rotates ~70˚, in response to aqueous conditions, taking two different conformations; the rising and resting P domain conformations. The P domain extends away from the S domain surface in solutions with higher pH, and rests on the S domain surface in solutions with lower pH. Metal ions help to stabilize the resting conformation in this

process. Furthermore, significant differences were found in the two P domain conformations with respect to MNoV infection of cultured cells. High-resolution cryo-EM structural analysis using MNoV-VLP revealed the structural similarity between infectious particle and VLP in MNoV and elucidates the molecular mechanism of the P domain rotation. Our findings provide new insights into the mechanisms of viral infection of the non-enveloped viruses.

## Results

### Dynamic rotation of the protruding domain in MNoV controls viral infection

For our structural studies, infectious particles of MNoV type 1 (MNoV-1) were produced by a reverse genetics system [20], propagated in RAW264.7 cells, and stored in DMEM (Dulbecco's Modified Eagle Medium). Viral particles were subjected to analysis by single-particle cryo-EM using a 200kV transmission electron microscope (TEM), and a 3D model was reconstructed at 5.3 Å resolution, in which the P domain was stabilized with an outer P2 subdomain-based interaction and rested on the S-domain (Fig 1A and 1C, S1A–S1D Fig). Interestingly, our results differed from the previous report of the 8 Å cryo-EM structure of the MNoV-1 suspended in PBS [14], in which the P domains rotated ~70˚ clockwise were mutually stabilized by interactions based on the inner P1 subdomain, and extended away from the S domain. To investigate these structural inconsistencies, we suspended the infectious particles in various aqueous solutions and examined them by cryo-EM. Finally, we were able to reproduce the reported MNoV-1 capsid structure at 7.3 Å resolution when the viral particles were suspended in a PBS(-) solution containing 20 mM EDTA at pH 8.0 (PBS(-)+EDTA (pH 8.0)) (Fig 1B, 1D and 1E, S1E–S1H Fig). The previously reported structure [14] of MNoV-1 was found to appear when pH was above 7 and metal ions were removed from solution by chelation with EDTA (Fig 1E). Furthermore, the dynamic rotation of the P domain of MNoV-1 infectious particles occurred reversibly in response to aqueous conditions, where the rotation from the rising to resting conformation was simply triggered at a pH lower than 7 (Fig 1E).

Next, we sought to elucidate the physiological functions of the two P domain conformations in terms of infectivity of the viral particles. When the MNoV-1 infectious particles pretreated with PBS(-)+EDTA (pH 8) were infected to RAW264.7 cells cultured in DMEM (pH 7.2–7.4) (see Materials and Methods), they showed significantly more propagation delay (~3 hours) than those pretreated with DMEM (Fig 1F). To investigate the reason for the propagation delay, we first compared the virus attachment on the host cell surface at 30 minutes after infection using quantitative real-time RT-PCR (qRT-PCR) (S2 Fig). Binding of the MNoV-1 particles pretreated with PBS(-)+EDTA (pH 8) to RAW264.7 cells, however, showed no significant difference from those pretreated with DMEM (Left bars in S2A Fig). RAW264.7 cells are known to express several molecules involved in MNoV adsorption [21]. In contrast, HEK293T cells genetically expressing CD300lf (HEK293T/CD300lf cells) can adsorb MNoV particles on the cell surface via direct interaction with the CD300lf [22, 23]. Thus, HEK293T/CD300lf cells were used to evaluate MNoV adsorption depending on the two P domain conformations. Using HEK293T/CD300lf cells, the MNoV-1 particles pretreated with PBS(-)+EDTA (pH 8) showed a significant reduction of cell binding compared to the particles pretreated with DMEM (Right bars in S2A Fig). In addition, flow cytometry (fluorescence-activated cell sorting (FACS)) analysis demonstrated that the MNoV-1 particles pretreated with PBS(-)+EDTA (pH 8) showed lower binding to cells (Orange in S2B Fig: Median = 268, CV = 157) than the MNoV-1 particles pretreated with DMEM (Light blue in S2B Fig: Median = 357, CV = 275), when the virus particles mixed with the cells at MOI (Multiplicity of infection) = 10. Furthermore, we compared early genome replication in RAW264.7 cells from 30 minutes to 12 hours

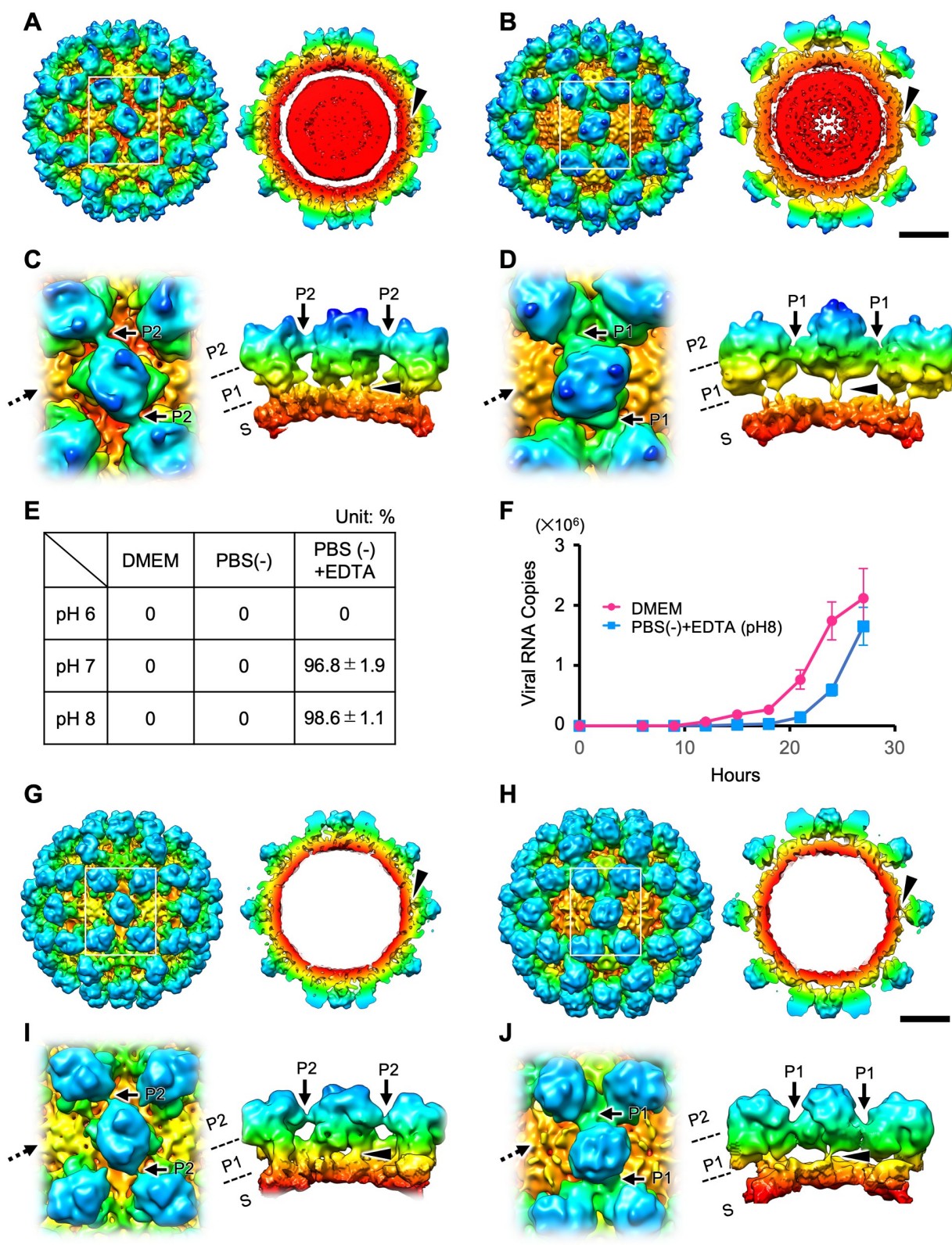

**Fig 1. Dynamic rotation of the P domain controls infection in norovirus.** (A and B) Cryo-EM structures of the MNoV-1 infectious particles suspended in DMEM (A) and PBS(-)+EDTA (pH 8) (B). The maps are low-pass filtered to 8 Å resolution to highlight the domain structure. The left and right panels are the isosurface representation and the center section, respectively. Black arrowheads indicate gaps between the S and P domains that appear in each of the resting and rising P domain conformations. (C and D) Left panels are the enlarged views of the boxes in A and B. Right panels are views from the direction of the dotted arrows shown in the left panel. The P domain of MNoV-1 suspended in DMEM rests on the S domain (black arrowheads in A and C) and interacts with the adjacent P domains at the outer P2 subdomain level (black arrows in C), called the resting P domain conformation. By contrast, the P domain of MNoV-1 suspended in PBS(-)+EDTA (pH 8) rises off the S domain (black arrowheads in B and D) and interacts with the adjacent P domains at the inner P1 subdomain level (black arrows in D), called the rising P domain conformation. In the rising P domain conformation, all P domain dimers rotate ~70˚ clockwise, compared to the resting P domain conformation, and the interaction sites are changed from the P2 to the P1 level. The P domain rising off from the S domain in PBS(-)+EDTA (pH 8) reversely rests on the S domain in DMEM. Scale bar, 100 Å. (E) Fraction of MNoV-1 particles having the rising P domain conformation in different aqueous conditions. Random sampling of 1,000 particles was carried out five times under each condition. The number of particles in each conformation was counted during image processing 2D classes. (F) Propagation curves of MNoV-1 infectious particles pretreated with DMEM and PBS(-)+EDTA (pH 8), respectively. MNoV-1 infectious particles were used to infect RAW264.7 cells and the number of the duplicated viral RNA copies along the timeline were measured by qRT-PCR. Error bars represent the standard deviations. MNoV-1 infectious particles pretreated with PBS(-)+EDTA (pH 8) propagated slower than those pretreated with DMEM. (G-J) Structures of the resting and rising P domain conformations in HNoV GII.3 VLPs. (G and H) Cryo-EM structures of the HNoV GII.3 VLP with the resting and rising P domain conformations (black arrowheads) displayed at 13 Å resolution. Isosurface and center section images of the whole particles are shown in left and right panels, respectively. Rotation of the P domain by ~70˚ and movement of the interaction site are similar to that of MNoV-1 (S4 Fig). Scale bar, 100 Å. (I and J) Left panels are enlarged views of the rectangle boxes in G and H. Right panels are views from the direction of the dotted arrows shown in the left panel. Black arrowheads indicate gaps between the S and P domains that appear in each of the resting and rising P domain conformations. Black arrows indicate the movement of the interaction site between the resting and rising P domain conformations.

(S2C Fig). The virus genome of the MNoV-1 pretreated with PBS(-)+EDTA (pH 8) increased only 3.6 fold, while that of the MNoV-1 pretreated with DMEM increased 26.4 fold. The results suggested that dynamic rotation of the P domain in MNoV-1 controls viral infection, where the resting P domain conformation induced by the DMEM pretreatment enhances binding to the target cells.

## Resting and rising P domain conformations in HNoV GII.3 VLP

We also observed the two conformations of the P domain in the HNoV GII.3 (TCH04-577 strain) VLPs by cryo-EM (S3 Fig). Single particle analysis showed that the two P domain conformations were similar to the resting and rising P domain conformations of MNoV-1 (Fig 1G–1J). In the HNoV GII.3 VLP suspended in DMEM, 16% of the total showed the rising P domain conformation, and the rest showed the resting P domain conformation. The ratio did not change even in PBS(-)+EDTA (pH 8.0).

The structure of the resting and rising P domain conformation was compared in more detail between MNoV-1 and HNoV GII.3 (S4 Fig). In the two structures, the angle of rotation between the resting and rising P domains was coincident (~70˚ clockwise), and the interactions between P domain dimers were similarly observed at the P2 levels in the resting conformation and at the P1 levels in the rising conformation, respectively. However, the heights of the P domain are significantly different in the rising P domain conformations, where the P domain of HNoV GII.3 is ~4 Å lower than MNoV-1. The length of the hinge loop between the S and P domains are the same in these NoVs (S5 Fig). It suggests that the holding manner in the hinge loop is different between MNoV and HNoV.

## Transformation process of the dynamic rotation of P domain in MNoV-1

We identified using cryo-EM that the rising P domain conformation gradually changed to the resting conformation in 2 to 8 hours, when MNoV-1 particles pretreated with PBS (-)+EDTA (pH 8) were suspended in DMEM (pH 7.2–7.4) (Fig 2A). Interestingly, particles mixed with two conformations were also observed between 2 and 6 hours (Fig 2B). The mixed P domain structure within a single particle suggests that even if the conversion of individual P domains may be rapid, it would take more time to modify the overall conformation of the capsid, where the P domains interact and are connected together like an entangled net, in which the P

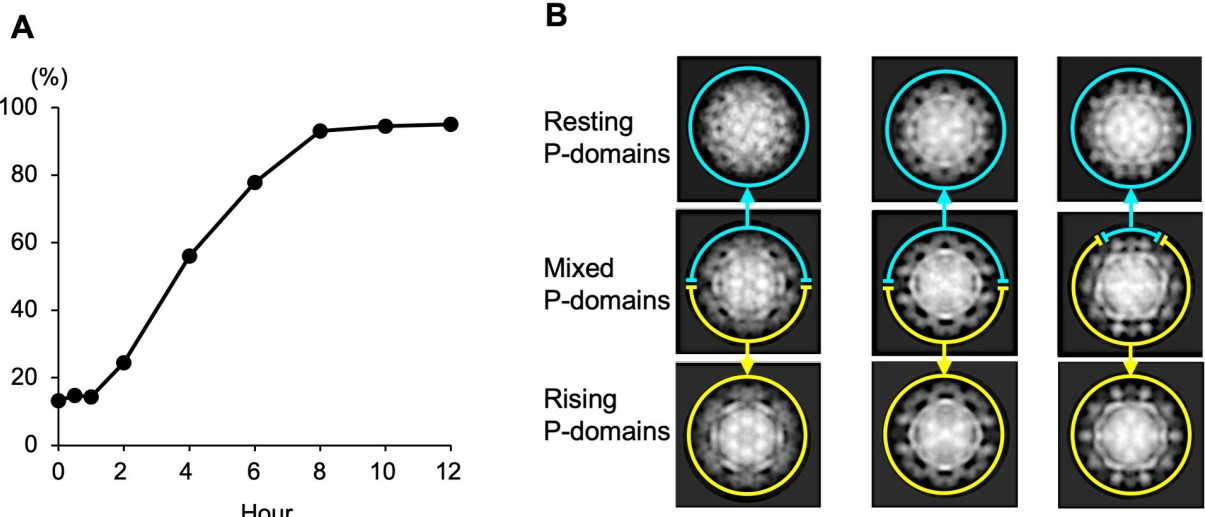

**Fig 2. Transformation process of the dynamic rotation of P domain in MNoV-1 infectious particles.** (A) MNoV-1 particles having the rising P domain conformation in PBS(-)+EDTA (pH 8) were suspended in DMEM. The particles were then observed directly using cryo-EM. The particles showing the resting conformation in all P-domains were counted in the classes after 2D classification. The percentage of the MNoV-1 particles having the resting P domain conformation were plotted over time (n = ~200 particles in each point). (B) Representative 2D average images of MNoV-1 particles between 2 and 6 hours. Particles mixed with two conformations were observed (second row panels). The mixed P domain structure within a single particle suggests that even if the conversion of individual P domains may be rapid, it would take time to modify the overall conformation of the capsid in this aqueous condition, where the P domains are connected together like an entangled net.

domain connections stabilize a given conformation causing hysteresis. In our observation, the transformations between the resting and rising P domain conformations were accelerated in higher or lower pH conditions, where they were completed in less than 5 minutes at the fastest, respectively. Consequently, the delay time of the virus propagation pretreated with PBS(-) +EDTA (pH 8) possibly corresponds to the time of the overall P domain conformational change of the virus particles in the cell culture media (pH 7.2–7.4). These results suggest that the rising conformation of the P domain prevents the initial viral binding to the host cell surface via CD300lf, and it takes time to recover infectivity by changing the rising conformation to the resting conformation in the whole capsid. Hence, MNoV-1 can control viral infectivity by the dynamic rotation of the P domain.

### Dynamic rotation of the P domain in MNoV-S7

The similar dynamic rotation of the P domain was also observed with MNoV type S7 (MNoV-S7) (S6 Fig). MNoV-S7 is a norovirus isolated from mouse stools in Japan in 2007 [24] and shows similar pathogenicity to MNoV-1. The amino acid sequence of MNoV-S7 VP1 protein shows 97% sequence identity with MNoV-1, with the nonhomologous 18 amino acids concentrated in the P domain (S5 Fig). Originally, we used this strain for structural study because we were already able to produce both the infectious particle and VLP [20]. However, since the P domain structure of the particle acquired was different from the previously reported cryo-EM map of MNoV-1 [14], MNoV-1 was used to confirm the structure in our sample preparation method. Interestingly, in the case of MNoV-S7, the rising P domain conformation only occurred in the solution of PBS(-)+EDTA at pH 8 or higher (S6K Fig), which is slightly higher than that of MNoV-1 (pH 7 or higher (Fig 1E)). The reversible rotation from

the rising to resting conformation was also triggered at a pH lower than 8 (S6K Fig). Viral propagation was delayed for the viruses pretreated with PBS(-)+EDTA (pH 8) solution compared to the viruses pretreated with DMEM, but the difference was smaller than that of MNoV-1 (Fig 1F and S6L Fig). The results indicate that the MNoV-S7 particles more readily convert conformation to the resting state than the MNoV-1 particles in the infection medium of DMEM (pH 7.2–7.4). However, binding of the MNoV-S7 particles pretreated with PBS(-) +EDTA (pH 8) to both RAW264.7 and HEK293T/CD300lf cells showed a significant difference from those pretreated with DMEM (S7A Fig). For early genome replication in RAW264.7 cells from 30 minutes to 12 hours, the virus genome of the MNoV-S7 pretreated with PBS(-) +EDTA (pH 8) increased only 7.4 fold, while that of the MNoV-S7 pretreated with DMEM increased 17.2 fold (S7B Fig). The binding mechanism of MNoV-S7 to host cells may be slightly different from that of MNoV-1. HNoV particles have often been shown to be unstable at alkaline pH, while sensitivity has been shown to vary significantly between genotypes and to be affected by slight sequence variations [25]. Differences in pH sensitivity between the two MNoV strains may also follow a similar mechanism.

## Capsid structure of MNoV-S7 VLP

To investigate molecular mechanism of the dynamic rotation of the P domain in the MNoV capsid, we determined the atomic structure of the entire MNoV capsid by single-particle cryo-EM using a 300kV TEM. VLPs of MNoV were used for this structural analysis, because the higher resolution capsid models are easier to obtain from VLPs than infectious particles containing nucleotides [26], and VLPs satisfy the biosafety level requirements of the 300kV EM room. VLPs of MNoV-1 and MNoV-S7 were individually prepared using a modified baculovirus expression system [27, 28] and stored in DMEM. The VLPs of MNoV-1 formed multiple types of icosahedral particles of 40–50 nm (S8 Fig), whereas those of MNoV-S7 formed a uniform icosahedral particle in the size of ~40 nm (S9A and S9B Figs). Therefore, the VLP of MNoV-S7 was used for the high resolution cryo-EM analysis, and the 3D model was determined at 3.5 Å resolution (Fig 3 and S9 Fig). Local resolutions were in the range of 3.3 to 3.7 Å in the capsid (Fig 3B). The highest resolution of 3.3 Å was observed in the S domain, and lower resolutions were mainly located in the peripheral regions of the P domain.

Atomic models of the MNoV-S7 VLP capsid were built for the quasi-equivalent A, B, and C monomers of VP1, respectively (S1 Table). The complete Cα backbone structure of the MNoV-S7 VLP is shown in Fig 3C. Other features of the models (e.g., S and P domains, Cα backbone of the VP1 C monomer) are presented in Fig 3D–3F. For modeling VP1's A and B monomers, 513 amino acids from Gln19 to Gly531 and 516 amino acids from Ala16 to Gly531 were used, respectively (S5 Fig). For modeling VP1's C monomers, 502 amino acids from Val30 to Gly531 were used (S5 Fig). The models were also fitted to the 5.2 and 5.3 Å resolution cryo-EM maps of the MNoV-S7 (S6E Fig) and MNoV-1 (S1C Fig) infectious particles with high cross-correlation coefficients of 0.92 and 0.91 (see Materials and Methods), respectively, indicating that the structure of MNoV-S7 VLP is highly similar to both the MNoV-S7 and MNoV-1 infectious particles.

A comparison of the atomic model of the MNoV-S7 VLP with the reported crystallographic models of the MNoV-1 P domain (PDB ID: 3LQ6 and 3LQE), which were expectedly analyzed in the aqueous condition of the rising P domain conformation [29], shows several interesting results. First, our cryo-EM model of the P domain in MNoV-S7, which was analyzed in the aqueous condition of the resting P domain conformation, had a higher degree of structural similarity to 3LQE than 3LQ6 (S10 Fig), suggesting that the cryo-EM model dominantly exhibits the closed state between the βA"–βB" loop and the βE"–βF" loop (S5 Fig, triple asterisks in

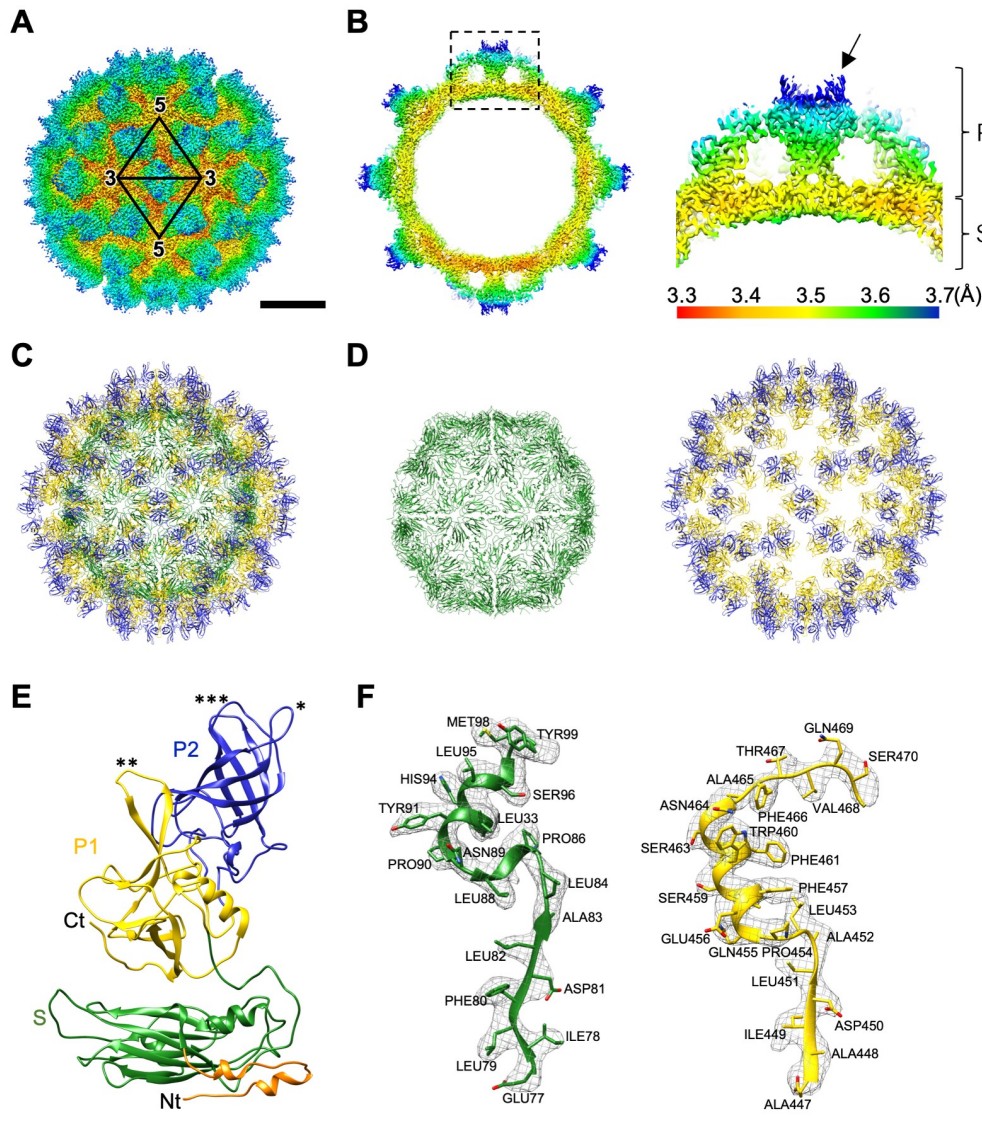

**Fig 3. Cryo-EM structure of MNoV-S7 VLP at 3.5 Å resolution.** (A) Cryo-EM map of MNoV-S7 VLP in DMEM viewed down the icosahedral twofold axis. Coloring is based on radii, as follows: red, up to 161 Å; yellow 161–171 Å; green, 171–191 Å; cyan, 191–211 Å; blue, 211 Å and above. Scale bar, 100 Å. (B) Local resolution assessment on a cross section of the MNoV-S7 VLP. The right panel shows a higher magnification of the square in the left image; the P and S domains are labeled. Coloring based on the local resolution at the bottom. Arrow indicates a horn-like structure in the P domain. (C and D) The Cα backbone of the S+P (C), S (the left panel in D), and P (the right panel in D) domains in the icosahedral MNoV-S7 particle at the same orientation with A. The coloring follows the standard designation of P1 subdomain (yellow), P2 subdomain (blue) and S domain (green). (E) A ribbon model of the *C* monomer of VP1 in the MNoV-S7 capsid. 502 amino acids from Val30 to Gly531were modeled in the C monomer, including the P1 subdomain (yellow), P2 subdomain (blue) and S domain (green). Single and double asterisks show the loops, which function to stabilize the P domain dimer. Triple asterisks show the βA"–βB" loop and βE"–βF" loop, which are in the closed state. (F) Representative cryo-EM electron densities of several amino acids and the fitted atomic models.

Fig 3E and S10 Fig). Second, the βC"–βD" loop (S5 Fig) extends upward with respect to the capsid surface (single asterisk in Fig 3E and single asterisk of magenta residues in S10 Fig) and forms an interaction with the βC'–βD' loop (S5 Fig, double asterisks in Fig 3E, and double

asterisks of magenta residues in S10 Fig) of the paired P domain in the P domain dimer in our cryo-EM model, whereas the βC''–βD'' loop (S5 Fig) in crystallographic models of MNoV-1 (3LQE than 3LQ6) extends downward with respect to the capsid surface (single asterisk of gray residues in S10 Fig) and form an interaction with the βC'–βD' loop (S5 Fig, double asterisks of gray residues in S10 Fig) of the paired P domain in the P domain dimer. Intriguingly, these two loops were disordered in the recent reported crystallographic model of the MNoV-1 P domain dimer containing the soluble domain of the cellular receptor, CD300lf (PDB ID: 5OR7) [30]. The facts suggest that an allosteric effect in the P domain [31] may exist during the dynamic P domain rotation and the receptor binding, though we have not had the detail structure of P domain in these conformations yet.

## Molecular interactions between VP1 proteins

To understand the capsid structure of the resting P domain conformation, which shows higher levels of infectivity, we investigated the molecular interactions between the P domain dimers of the MNoV-S7 VLP suspended in DMEM. The cryo-EM map of the MNoV-S7 VLP in DMEM revealed several hydrophobic and polar interactions between the neighboring A/B and C/C dimers at the P2 level (yellow in Fig 4). We propose the following chemical interactions between residues on the P domain surface: Asn409, Gly411, Leu412, and Pro415 located on the βF''–βB' loop (S5 Fig) of the C/C dimer interact with Gln371, Arg373, Val368, and Pro319 located on the βB''–βC'' and βD''–βE loops (S5 Fig) of the A/B dimers, respectively (Fig 4B and 4C).

As expected from sequence homology (S5 Fig), the structure of the S domain of MNoVs was similar to the HNoV GI.1 VLP (PDB ID: 1IHM) [7] except for some local distortions. Our cryo-EM structural study showed several elaborate interactions between adjacent N-terminals of the S domains, by successfully modeling the residues from Gln19 in the A monomer, Ala16 in the B monomer, and Val30 in the C monomer (S11A Fig). The flexible N-terminals extended from A, B, and C monomers formed various interactions with each other between adjacent S domains at the twofold, threefold, pseudo-threefold, and fivefold axes, respectively (S11B–S11F Fig). A new feature found in the structure of MNoV is that the N-terminal arm (NTA) in the A monomer extends to the adjacent A monomer, forming a complex of the fivefold axis (S11E Fig). On the other hand, the NTA in the B monomer extends to the adjacent C monomer, forming a complex of the threefold axis (S11F Fig), though the NTA in C monomer is more disordered. Eventually, these NTAs contribute to form the stable icosahedral inner shell.

As shown in S11G–S11H Fig, the previously reported NTAs of caliciviruses are structurally divided into two groups. One group, belonging to norovirus and lagovirus shown in HNoV GI.1 [7] and RHDV [15], is that the NTA runs along the bottom edge of its S domain and interacts with the adjacent S domains (blue and cyan in S11G and S11H Fig). The other group, belonging to vesivirus shown in SMSV [16] and FCV [17] is that the NTA runs along the bottom edge of the adjacent S domain and interacts with the adjacent S domains (yellow and green in S11G and S11H Fig). NTA in MNoV has a similar conformation as the first group (red in S11G and S11H Fig), further clarifying the function of the NTA in the A monomer. Consequently, the rigid capsid shell of MNoV was maintained by these outer P domain interactions and inner S domain crosslinks.

## Molecular mechanism of the reversible rotation of the P domain dimers

We investigated the capsid structure of MNoV in the rising P domain conformation to understand the molecular mechanism of the dynamic rotation of the P domain. The atomic model

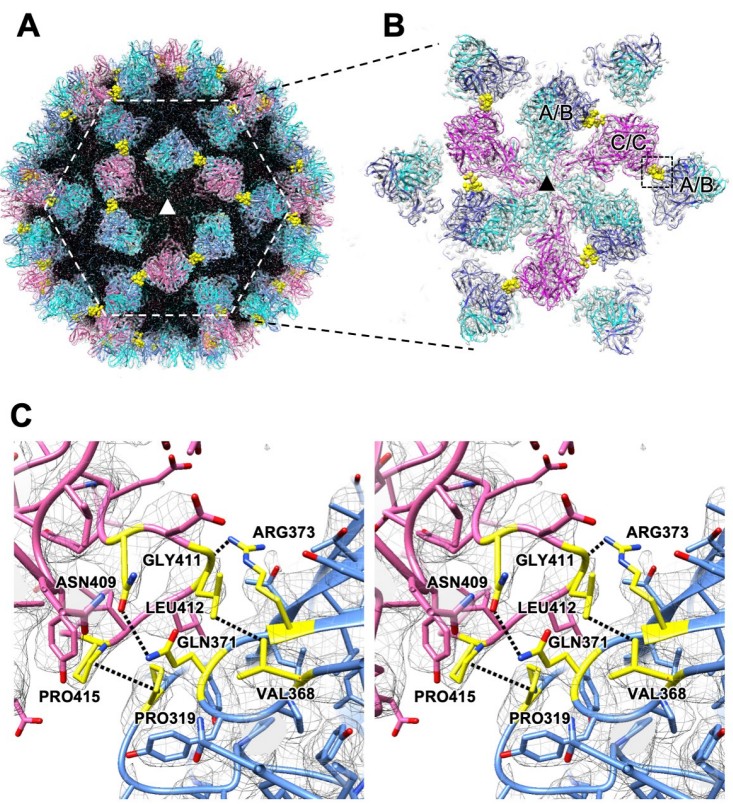

**Fig 4. Interactions between the resting P domain dimers.** (A) A cryo-EM map of the MNoV-S7 VLP highlights interactions of the P domain dimers. The central threefold axis is labeled with a triangle. To emphasize the P-domains, the density map corresponding to the S-domain is labeled in black. (B) The enlarged view of the dotted hexagon in A. C/C-dimers (purple) on the twofold axis interact with adjacent A/B-dimers (blue) located around the fivefold axis. The residues interacting with the adjacent P domain dimers are colored yellow. (C) A stereo view of the candidate chemical interactions between the C/C and A/B dimers indicated by the dotted square in B. Asn409, Gly411, Leu412, and Pro415 of the C/C dimer are facing with Gln371, Arg373, Val368, and Pro319 of the A/B dimer, respectively (S5 Fig), where hydrophobic and polar interactions are suggested to be formed.

of the MNoV-S7 VLP in DMEM was modified and fitted into the 7.2 Å cryo-EM map of the MNoV-S7 infectious particle suspended in PBS(-)+EDTA (pH 8) with the high cross-correlation coefficient of 0.92 (S6I and S6J and S12A Figs) (see Materials and Methods). The map to model Fourier Shell Correlation (FSC) showing a similar curve with "gold standard"-FSC (GS-FSC) indicated 8.5 Å resolution at 0.5 criteria (S12B Fig). This suggests that the atomic model of the resting P domain conformation is very similar to the rising P domain conformation at the resolution. The model in PBS(-)+EDTA (pH 8) showed that the P domain rotates clockwise by ~70˚ and rises up from the S domain surface by ~13 Å compared to the original model in DMEM (Fig 5 and S12A, S13A and S13B Fig). The atomic model fitted to the cryo-EM map of the rising conformation suggested that a new interaction links the P1 subdomain of the C/C dimer to the adjacent P1 subdomain of the A/B dimers via hydrophobic and polar interactions (S13C Fig). These are formed between residues on the newly facing P domain surface; i.e., Pro425, Ser504, Leu524, and Gln526 in the βF"–ββ' and βF'–αG loops, while the C-terminal of the P1 subdomain in the C/C dimer interact with Phe423, Gln263, Leu524, and Arg523 located in the same loops and C-terminal in the A/B dimer, respectively (S5 Fig).

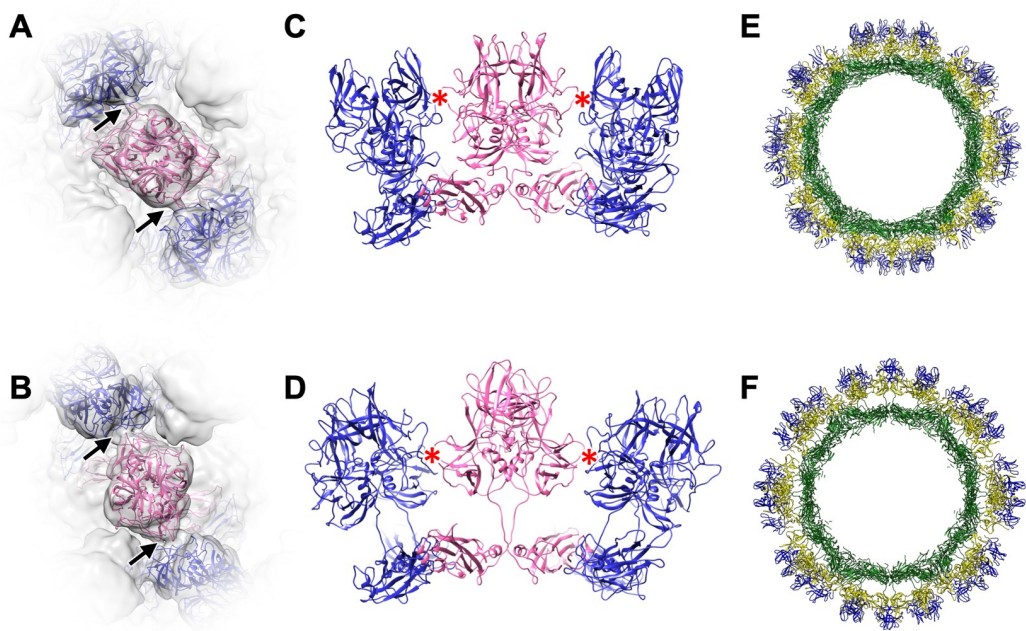

**Fig 5. Molecular mechanism of the reversible dynamic rotation of the P domain dimers.** (A and B) Ribbon models of the C/C dimers of MNoV-S7 in DMEM and PBS(-)+EDTA (pH 8), which are modified and fitted into each cryo-EM map. Arrows indicate where the C/C dimer interacts with the adjacent A/B dimers. (C and D) Side views of the ribbon models showed in A and B, respectively. The interactions linking the adjacent P domains are indicated by red asterisks. The animated dynamic rotation of the P domain dimers between two conformations are shown in S1 Movie. (E and F) Central slices of the MNoV particle in the resting and rising P domain conformations.

An animated comparison of the P domain conformations and reversible structural dynamics of the MNoV capsid under different aqueous conditions is shown in S1 Movie. The P domains in DMEM rested on the S domains and were stabilized by interaction between the adjacent P domains at the P2 level (Fig 5A, 5C and 5E), while the P domains in PBS(-)+EDTA (pH 8) rose up from the S domain and were stabilized by interaction between the adjacent P domains at the P1 level (Fig 5B, 5D and 5F). The flexibility of the hinge connecting the P and S domains (S14A Fig) is important for dynamic twisting of the P domain, which moves the P domain up and down by ~13 Å. In addition, as the P domain rose, the A/B dimer was bent slightly toward the fivefold axis (S15 Fig).

## Discussion

Here, we reported that reversible dynamic rotation of the P domain results in two conformations of the MNoV capsid. The two conformations of P domain were also identified in HNoV GII.3 VLP at first time. So far, only one of two conformations has been reported in each HNoV VLP (resting conformation: GI.1 (pH4.8), GI.7, GII.2; rising conformation: GII.4, GII.10) and other caliciviruses [11,13,32]. In the case of MNoV, the main trigger for the conformational change was pH of the solution: a higher pH changed the conformation of the P domain to the rising state, while a lower pH changed it to the resting state. Interestingly, the pH thresholds differed by genotype: MNoV-1 showed pH 7 (Fig 1E) and MNoV-S7 showed pH 8 (S6K Fig). However, it was not clear what determines the threshold for interconversion. Some different amino acids localized between the P and S domains may function by altering

the surface charge, while metal ions bound to the region between the P and S domains, may also help to stabilize the resting conformation. Therefore, in addition to higher pH, EDTA is necessary to remove metal ions from solution, permitting the conversion of the capsid protein to the rising state. As a candidate, we found a potential interaction via a metal ion between two carboxylates (Glu223 and Asp231) observed in the folded, flexible hinge of the resting conformation (S14 Fig). This suggests that the flexible hinge between the P and S domains could be folded by binding metal ions in DMEM and unfolded by releasing the metal ion in PBS(-) +EDTA (pH 8). In addition, several potential interaction sites connecting the S and P domains are also identified in the resting state map (e.g. Glu66 in S domain—Ser519 in P domain, Glu176 in S domain—Tyr227 in P domain, Asp174 in S domain—Gln469 in P domain) (S5 Fig). These residues may function to stabilize the resting conformation via metal ions in DMEM, which includes many inorganic salts such as $CaCl_2$, $(Fe(NO_3)_3 \cdot 9H_2O$, $MgSO_4$, KCl, $NaHCO_3$, NaCl, $NaH_2PO_4$-$H_2O$ [33]. At present, the metal ions that contribute to domain binding have not been identified, so that the dynamic P domain rotation cannot be controlled by changing their concentrations. To fully understand this mechanism, it is necessary to identify the localization of all metal ions that directly couple the P and S domains must be determined in the future.

Our observations suggest that the capsid of HNoV can also adopt two reversible P-domain conformations, though the dynamic rotation is currently identified only in the MNoV particles. Further investigations are needed to determine the factors that control the orientation of the P domain in HNoV. So far, the resting and rising conformations were independently observed in several HNoV VLPs [32]. However, this report is the first case to present two conformations simultaneously in the same particle in a reversible state. Koromyslova et al. reported the structural alterations in presence of citrate in HNoV GII.10 VLP [34], which strongly resemble the micrographs of the rising P domain in S1E and S6G Figs. We, however, believe the mechanism works differently in the expanded VLP in citrate, because we have confirmed the GII.10 VLP originally represented the rising conformation in the aqueous condition and the resting conformation has not yet been identified [13].

We observed that MNoV-1 particles showed slow transformation between the resting and rising P domain conformations in near neutral pH solution. In HNoV GII.4, post-translational modification of Asp373 by a deamidation was reported to attenuate histo-blood group antigen (HBGA) binding with an estimated half-life of a few days [35]. The slow reaction is also strongly linked to pH. However, it seems to be a different mechanism from the P-domain rotation in MNoV, as no compatible residues of Asp373 were found in the sequences (S5 Fig), in addition to the fact that glycan binding is not essential for infection of MNoV. The slow regulation step to control infectivity may be common between MNoV and HNoV and essential for infection in the dramatic environmental changes present in the gastrointestinal tract, though the approaches are different.

Our infection and adhesion experiments of the virus particles to the host cells clearly suggested that the viruses having the resting P domain conformation attach to and infect host cells more easily than the viruses having the rising P domain conformation. To structurally confirm the difference in the affinity for the host cell surface between the resting and rising P domain conformations, the crystallographic model of the P domain and soluble domain of CD300lf (sCD300lf) (PDB ID: 5OR7) was fitted to the cryo-EM maps of MNoV-1 in the different P domain conformations. As shown in S16 Fig, the sCD300lf binding site on the P domain is formed inside a triangle between one C/C dimer and two A/B dimers on the viral surface. In the resting P domain conformation, the triangle is distorted and the sCD300lf molecules are more flexibly able to approach the binding site (S16A Fig). On the other hand, in the rising P domain conformation, the triangle is squeezed, limiting the space for the sCD300lf molecules

to approach the binding site (S16B Fig). The previous reports also suggested that a steric hindrance in the sCD300lf molecules would occur between the P domain dimers in the rising P domain conformation, where the C-terminals of sCD300lf clashed in its multiple conformation conforming to the space formed between the P domain dimers, while it had more space between the P domain dimers in the resting conformation [11,19,30,31]. The adaptation of the sCD300lf binding model in our cryo-EM map clearly showed that the accessibility of CD300lf to the P domain is altered in the resting and rising P domain conformations in the single virus. It shows a possibility that the rotation of the P domain modulates viral infectivity by altering the accessibility of the viral epitopes to cellular receptors.

In consequence, the resting P domain conformation is more accessible to the cellular receptor CD300lf than the rising P domain conformation, causing greater viral infectivity. However, it raises a question: why does norovirus require such a reversible rising P domain conformation? One possibility is that the conformational changes are important during virus assembly and disassembly in host cells. Tobacco mosaic virus (TMV), well-studied for its structural and physiological characteristics, forms a metastable structure in the cell, where lower concentration of metal ions and higher pH are proposed as the trigger for virus disassembly [36]. A recent cryo-EM study showed the calcium ion-dependent conformational changes of TMV, indicating that chemical bonds involving calcium ions prevents disassembly of TMV [37]. Dengue and Zika viruses also show pH-dependent structural changes associated with viral replication [38,39]. In this study, we observed the MNoVs having the rising P domain conformation showed about 2 Å lower resolution (~7.2 Å) than the MNoVs having the resting P domain conformation (~5.2 Å) (S1D, S1H, S6F and S6J Figs). In addition, the particles finally contributed to the 3D map were 23~28% of the initially picked particles in the rising conformation data set, while they were 91~95% in the resting conformation data set (S2 Table), where mixed P domain conformations, departure from icosahedral symmetry, and broken particles were removed during iterative 2D and 3D map refinements. These facts suggest that MNoV with the rising P domain conformation are relatively flexible and fragile. Such structural flexibility and fragility may facilitate viral assembly and disassembly. The flexibility and fragility of the rising P domain conformation was also identified in HNoV GII.3 VLPs, where the rising P domain conformation easily collapsed in thinner vitreous ice film due to surface tension (S17C Fig), and thus the only particles with the resting P domain conformation were left in the central thinner ice area of the cryo-EM grid hole (S17A and S17B Fig). In the case of HNoV GII.3 VLPs, the particles finally contributed to the 3D map was 29% of the initially collected particles in the rising conformation data set, while they were 50% in the resting conformation data set (S2 Table). Assuming that the rising P domain conformation represents an intermediate state of the viral assembly and disassembly, norovirus changes its structure to a metastable form upon entry into host cells, and release the viral genome into the cytosol at the relatively lower metal ion concentration and higher pH environment of the cytoplasm. Conversely, norovirus preassembled in host cells may be further modified to its final stable form in the more demanding extracellular environment, such as the gastrointestinal tract, which is rich in free metal ions and lower pH. Another hypothesis has been suggested that the fragile and flexible P domains function in antibody escape or facilitate binding to target cells [11]. We have to wait for future study to provide an answer to this question.

Recently, a dynamic feature of calicivirus capsid was reported in FCV. There, the twelve copies of the minor capsid protein, VP2, form a portal-like assembly on the capsid surface after receptor engagement, suggesting that it functions as a channel for delivery of the viral genome [18]. As with FCV, no VP2 density was found in our symmetry-imposed cryo-EM map of noroviruses. One reason is that the copy number is limited and does not match the icosahedral symmetry, where the components disappear due to imposition of symmetry.

Assuming the same mechanism as FCV, VP2s in noroviruses dispersed inside the particle could accumulate and be assembled according to the structural changes due to receptor involvement. It is unknown whether MNoV has a similar portal structure consisting of VP2 at present, but a rearrangement of the capsid structure, such as the dynamic rotation of the P domain in addition to the receptor binding is believed to be a necessary step to form the portal and release the viral genome.

Interactions between P domain dimers on the capsid surface have also been observed in human noroviruses, but are more complex and diverse. The HNoV GI.1 VLP shows dimer-dimer interactions at the P2 subdomain level, where the P2 subdomain of the C/C dimer links with all four adjacent A/B dimers and the P domains rest on the S domains [7]. By contrast, in HNoV GII.10 VLP the similar dimer-dimer interactions are carried out at the P1 subdomain level not at the relatively small P2 subdomains, and the P domain rises from the S domain by ~16 Å, like the rising P domain conformation in MNoV. These different conformations of the P domains in the two HNoV strains result from switching interaction sites at the P1 or P2 levels, which can be structurally achieved by a simple rotation of the P domain. In addition, we previously reported a unique neutralizing antibody against the HNoV GII.10 VLP, which binds to the site of occlusion between the P and S domains of the viral capsid, inhibiting binding to target cells [13]. This suggests that the antibody inhibits viral-cell attachment or/and infection by interfering with the rotation of the P domain from the rising position to the resting position. Currently, dynamic rotation of the P domain has been identified only in the infectious particles of MNoV. Further studies will reveal the full mechanism of the non-enveloped calicivirus infection.

## Materials and methods

### Sample preparations of norovirus infectious particles and VLPs

Infectious particles of MNoV-S7 were produced using plasmid-based reverse genetics as previously described [22]. Those of MNoV-1 were produced in the same procedure as MNoV-S7. pMuNoV-MNoV-1 was constructed by exchanging the MNoV sequence portion from MNoV-S7 to MNoV-1 with the In-Fusion cloning system (Takara Bio Inc.), according to the manufacturer's protocol. MNoV-1 infectious cDNA was kindly provided by I. Goodfellow as an MNoV-1 cDNA plasmid [40]. Infectious viruses of MNoV-1 and MNoV-S7 were propagated with RAW264.7 cells (American Type Culture Collection). The culture supernatant was collected two days after infection. MNoV was pelleted using ultracentrifugation with a 30% sucrose cushion and purified with cesium chloride (CsCl) equilibrium ultracentrifugation. MNoV was diluted with DMEM without any additional supplements and pelleted again to remove CsCl. The purified MNoV pellet was resuspend in DMEM (Nacalai Tesque Inc.) or phosphate-buffered saline without calcium and magnesium containing 20mM EDTA at pH 8.0 (PBS(-)+EDTA (pH 8.0)) to prepare $10^7$ infectious virions /µL. VLPs of MNoV-1, MNoV-S7, and HNoV GII.3 TCH04-577 strains were produced by a baculovirus expression system [27,28] and purified with CsCl equilibrium ultracentrifugation in the same procedure as infectious particles.

### Analysis of MNoV one-step propagation curve

RAW264.7 cells were cultured at a density of $10^6$ cells/well in 48-well plates and infected with the purified infectious particles of MNoV-1 or MNoV-S7 at a MOI = 2 or more. After 30 min of incubation at 37˚C, the inoculum was removed, and the cells were washed three times with DMEM to remove unbound viruses, and DMEM containing 10% FBS (Thermo Fisher Scientific) was added. The plates were incubated at 37˚C for the stated time; time zero indicates the

time at which the medium was added. After 0, 3, 6, 9, 12, 15, 18 and 24 h of incubation, 20 μL of the culture medium was sampled and centrifuged at 10,000 g for 10 min at 4˚C to remove cells and cell debris. 15 μL of supernatant was collected and stored at -80˚C until RNA extraction. 10 μL of the supernatant was used for RNA extraction and qRT-PCR, according to published protocols [41]. These assays were performed three times independently and calculated standard deviations (SD) plotted at each point.

### Evaluation of MNoV-1 Binding using FACS

MNoV-1 binding to the host cells was examined by FACS (fluorescence activated cell sorter) analysis. Briefly, $1 \times 10^6$ of HEK293T/CD300lf cells were incubated with purified infectious MNoV-1 particles ($1 \times 10^9$ CCID50), that were suspended in DMEM-10% FBS or PBS (-) + 20mM EDTA (pH 8), for 30 min at 4˚C. After washing, the cells were incubated with anti-MNoV VP1 rabbit polyclonal antibody labeled by Dylight 488 (LNK221D488, BioRad) for 30 min at 4˚C. The solution in each step includes 3% FCS and 20 mM $NaN_3$ to prevent the cells from virus uptake. Afterward, the cells were washed again and analyzed by BD FACSMelody (BD Bioscience) and analyzed by using FlowJo software.

### Evaluation of cell binding and early genome replication of MNoV in the host cells using qRT-PCR

RAW264.7 or HEK293T/CD300lf cells were cultured at a density of $10^5$ cells/well in 96-well plates and infected with the purified infectious particles of MNoV-1 or MNoV-S7 at a MOI ≦ 10. After 30 min of incubation at 37˚C, the cells were washed twice with DMEM to remove unbound viruses. The total RNA was extracted by NucleoSpin RNA (Takara Bio Inc.) and the virus particles attached on the cell surface were estimated by qRT-PCR, according to published protocols [41]. The early genome replication in each cell incubated for 0, 2, 4, 6, 8, 10, and 12 hours post-infection. RNA was extracted using the same procedure as above and used for qRT-PCR. These assays were performed three times independently and calculated standard deviations (SD) values and plotted at each point.

### Cryo-EM and image processing

Aliquots (2.5 μL) of the purified MNoV infectious particles and HNoV GII.3 VLP were placed onto R 1.2/1.3 Quantifoil grids (Quantifoil Micro Tools) coated with a thin carbon membrane that were glow-discharged using a plasma ion bombarder (PIB-10, Vacuum Device Inc.) immediately beforehand. These grids were then blotted and plunge-frozen using a Vitrobot Mark IV (Thermo Fisher Scientific) with the setting of 95% humidity and 4˚C. Vitreous ice sample grids were maintained at liquid-nitrogen temperature within a JEM2200FS electron microscope (JEOL Inc.), using a side-entry Gatan 626 cryo-specimen holder (Gatan Inc.), and were imaged using a field-emission gun operated at 200 kV and an in-column (Omega-type) energy filter operating in zero-energy-loss mode with a slit width of 20 eV. Images of the frozen hydrated norovirus particles were recorded on a direct-detector CMOS camera (DE20, Direct Electron, LP) at a nominal magnification of 40,000×, corresponding to 1.422Å per pixel on the specimen. Using a low-dose method, the total electron dose for the specimen is about 20 electrons per $Å^2$ for a 3-second exposure. Individual images were subjected to per-frame drift correction by a manufacturer provided script.

For MNoV-1 infectious particles suspended in DMEM and PBS(-)+EDTA (pH 8.0), 6,708 and 4,704 particles were selected from 1,046 and 2,188 images, respectively, and then extracted using RELION 2.0 [42] after determining the contrast transfer function (CTF) with CTFFIND4 [43]. Alignment and classification of extracted particles were performed, and a 3D

map was reconstructed in RELION 2.0 by using an initial model that generated with icosahedral symmetry by EMAN1 [44]. The 3D reconstructions were computed, and the final resolutions of the density maps were estimated to be resolutions of 5.3 and 7.3 Å, respectively, using a GS-FSC (cutoff 0.143) between two different independently generated reconstructions [45]. 3D renders of the maps were created in UCSF Chimera [46]. For MNoV-S7 infectious particles suspended in DMEM and PBS(-)+EDTA (pH 8.0), 2,049 and 2,739 images were collected, respectively, and the 3D reconstructions were calculated with 17,820 and 5,063 particles at 5.2 and 7.2 Å resolution, respectively. For HNoV GII.3 VLP containing the resting P domain and the rising P domain, 106 and 1,917 images were collected, and 3D reconstructions were calculated with 279 and 1,482 particles at 9.3 and 12.9 Å resolution, respectively. Data collection and image processing are summarized in S2 Table.

For high-resolution structural analysis, cryo-EM images for MNoV-S7 VLP were acquired with a Falcon II detector at a nominal magnification of 75,000×, corresponding to 0.86 Å per pixel on a Titan Krios at 300 kV (Thermo Fisher Scientific). A low-dose method (exposures at 20 electrons per Å$^2$ per second) was used, and the total number of electrons accumulated on the sample was ~40 electrons per Å$^2$ for an 2 second exposure. A GIF-quantum energy filter (Gatan Inc.) was used with a slit width of 20 eV to remove inelastically scattered electrons. Individual micrograph movies were subjected to per-frame drift correction by MotionCor2 [47]. Particles were selected from the 2,746 images and the final 3D reconstruction was computed with 41,847 particles. The resolution of the density map was estimated to be 3.5 Å using a GS-FSC criterion. Data collection and image processing are summarized in S1 Table.

## Atomic model building of MNoV-S7 VP1

The P domain atomic model of MNoV-1 (PDB ID: 3LQE) and the S-domain atomic model of HNoV (PDB ID: 1IHM) were used as templates for the homology model building of the P and S domains of the MNoV-S7 VP1, respectively. Multiple-sequence alignments of VP1s of MNoV-S7, MNoV-1 and HNoV were performed using the PROMALS3D program [48]. The sequence alignments of the P and S domains were used as the input of MODELLER software [49] to generate the comparative models. The maps containing A, B, and C monomers were extracted using UCSF Chimera [46], respectively, for VP1s of MNoV-S7 and MNoV-1. The models were manually re-built in the individual maps from the homology model mentioned above using COOT [50] and refined using PHENIX [51]. The models fitted to the cryo-EM maps were evaluated with the cross-correlation coefficient between the model and the cryo-EM map respectively, using "Fit in Map" of UCSF Chimera [46]. Data collection, image processing, and model statistics are summarized in S1 Table.

## Supporting information

**S1 Table. Data collection, image processing and model statistics (MNoV-S7 VLP)** (DOCX)

**S2 Table. Data collection and image processing (MNoV infectious particles and HNoV GII.3 VLP)** (DOCX)

**S1 Fig. Cryo-EM map generation of the MNoV-1 infectious particles suspended in DMEM and PBS(-)+EDTA (pH 8).** (A and E) Representative micrographs of MNoV-1 in DMEM and PBS(-)+EDTA (pH 8). Scale bars, 500 Å. (B and F) The representative 2D class average images derived from A and E, respectively. Scale bars, 200 Å. (C and G) The surface-shaded depth-cued representations of the MNoV-1 in DMEM and PBS(-)+EDTA (pH 8) viewed along the

icosahedral twofold axis. Coloring is based on radii, as in Fig 1A. Scale bars, 100 Å. (D and H) GS-FSC plots of the cryo-EM maps of MNoV-1. Based on the 0.143 criterion for comparing two independent data sets, the image resolutions are estimated to be 5.3 and 7.3 Å, respectively.
(TIF)

**S2 Fig. Viral adhesion of MNoV-1 particles to the host cells.** (A) Viral initial attachment on the host cells (RAW264.7 or HEK293T/CD300lf) at 30 minutes after post-infection of the MNoV-1 particles pretreated with DMEM or PBS(-)+EDTA (pH 8). After washing the cells with the solution, the viral RNA was extracted from viruses attached on the cells and the amount was estimated by qRT-PCR. *(p<0.05). (B) The HEK293T/CD300lf cells were incubated with the MNoV-1 particles pretreated with DMEM (Orange), the MNoV-1 particles with PBS(-)+EDTA (pH 8) (Light blue), or without MNoV-1 particles (Red dots). MOI was 10. After incubation, these cells were immunostained with anti-MNoV VP1 antibody. X-axis represents the range of the MNoV binding, and y-axis represents the number of the cells. Median and CV values are 357 and 275 (MNoV-1 particles pretreated with DMEM), and 268 and 157 (MNoV-1 particles pretreated with PBS(-)+EDTA (pH 8)), respectively. (C) Early genome replication of MNoV-1 from 30 minutes to 12 hours. RAW264.7 cells infected with the MNoV-1 particles pretreated with DMEM or PBS(-)+EDTA (pH 8) were washed and collected at each time point of post-infection. The RNA was extracted and the amount was estimated by qRT-PCR.
(TIF)

**S3 Fig. Cryo-EM map generation of the HNoV GII.3 VLPs.** (A and B) Representative 2D class average images of the HNoV GII.3 VLPs with the resting and rising P domain conformations. Scale bars, 200 Å. (C) A surface-shaded depth-cued representation of the HNoV GII.3 VLP with the resting P domain at 9.3 Å resolution. Viewed down the icosahedral twofold axis. Coloring is based on radii as in Fig 1A. Scale bar, 100 Å. (D) GS-FSC plots of the cryo-EM maps of the HNoV GII.3 VLP with the resting and rising P domain conformations indicated by black and gray lines. Based on the 0.143 criterion for comparing two independent data sets, the resolutions of the reconstruction are 9.3 and 13 Å, respectively.
(TIF)

**S4 Fig. Comparison of the capsid shell between MNoV-1 and HNoV GII.3.** (A) Images of Fig 1C (MNoV-1 with the resting P domain conformation) and Fig 1I (HNoV GII.3 with the resting P domain conformation) are compared side by side. (B) Images of Fig 1D (MNoV-1 with the resting P domain conformation) and Fig 1J (HNoV GII.3 with the resting P domain conformation) are compared side by side. Interaction points between C/C dimer and A/B dimers are labeled by black arrows. In the both strains, the rotation angles between the resting and rising P domains are ~70° clockwise, and the interactions between P domains were also observed similarly at the P2 levels in the resting conformation and at the P1 levels in the rising conformation. However, the heights of the P domain are significantly different in the rising P domain conformations, where the height of P the domain in HNoV GII.3 is ~4 Å shorter than MNoV-1. The P domain structure of MNoV-S7 was similar to that of MNoV-1 at this resolution (in main text).
(TIF)

**S5 Fig. Sequence alignment of the VP1 protein in HNoV(GI.1, GII.3), MNoV-1, and MNoV-S7.** For comparison, the amino acid sequences of the VP1 of HNoV(GI.1, GII.3) and MNoV-1 are aligned to that of MNoV-S7. Crystallographic structures of the HNoV GI.1 S (PDB ID: 1IHM) and the MNoV-1 P (3LQE) domains were used for the initial homology

model building of MNoV-S7. Orange, green, yellow, and blue arrows above the alignments represent the N-terminal, S domain, P1 subdomain, and P2 subdomain regions, respectively. Conserved sequences (46%) among HNoV and MNoV are indicated by gray, and different sequences (6%) between MNoV-1 and MNoV-S7 are indicated by red. Regions corresponding to the major β-strands in norovirus capsid are represented by black arrows. Arrowheads indicate start and end residues of the A, B, and C monomers to be built MNoV-S7 structure model, respectively. MNoV-S7 s.s. indicates the predicted secondary structures.
(TIF)

**S6 Fig. Dynamic rotation of the P domains controls viral infection in MNoV-S7.** (A and B) Cryo-EM structures of the MNoV-S7 infectious particles, suspended in DMEM and PBS(-) +EDTA (pH 8) are low pass filtered to 8 Å resolution to highlight the P domain structure. Left and right panels show the isosurface display and center section, respectively. The P domain of MNoV-S7 suspended in DMEM rests on the S domain (arrow in A) and interacts with the adjacent P domains at the outer P2 subdomain level (asterisks in A), called the resting P domain conformation. By contrast, the P domain of MNoV-S7 suspended in PBS(-)+EDTA (pH 8) rises off the S domain (arrow in B) and interacts with the adjacent P domains at the inner P1 subdomain level (asterisks in B), called the rising P domain conformation. In the rising P domain conformation, all P domain dimers rotate ~70° clockwise, compared to the resting P domain conformation (highlighted by white boxes), and the interaction sites are changed from the P2 to the P1 levels. The P domain rising off from the S domain in PBS(-) +EDTA (pH 8) reversely rests on the S domain in DMEM. Scale bar, 100 Å. (C and G) Representative micrographs of MNoV-S7 in DMEM and PBS(-)+EDTA (pH 8). Scale bars, 500 Å. (D and H) The representative 2D class average images derived from C and G. Scale bars, 200Å. (E and I) The surface-shaded depth-cued representations of MNoV-S7 in DMEM and PBS(-) +EDTA (pH 8) viewed down the icosahedral twofold axis, respectively. Coloring is based on radii, as in Fig 1A. Scale bar 100 Å. (F and J) GS-FSC Plots of the cryo-EM maps of MNoV-S7 (E and I). Based on the 0.143 criterion for comparing two independent data sets, the resolutions of the reconstruction are 5.2 and 7.3 Å, respectively. (K) The fraction of MNoV-S7 particles exhibiting the rising P domain conformation in different aqueous conditions. Random sampling of 1,000 particles was carried out five times under each condition. The number of particles in each conformation was counted from the individual 2D classes. The rising P domain conformation of MNoV-S7 appeared at slightly higher pH than MNoV-1 (Fig 1E). (L) Propagation curves of MNoV-S7 pretreated in DMEM and PBS(-)+EDTA (pH 8), respectively. The infectious particles were infected to RAW264.7 cells and measured the number of the duplicated viral RNA copies along the timeline by qRT-PCR. Error bars represent the standard deviations. The values of the data sets in DMEM and PBS(-)+EDTA (pH 8) were slightly closer and the Mann-Whitney rank test was performed at each time point as follows: 0h 0.512; 6h 0.827; 9h 1.997; 12h 0.335; 15h 0.049; 18h 0.049; 21h 0.049; 24h 0.335. Significant differences ($p < 0.05$) between the two data set were found after 15 hours. MNoV-S7 pretreated with PBS (-)+EDTA (pH 8) produced slower propagation of the virus than that pretreated with DMEM, though the difference was small compared to that of MNoV-1 (Fig 1F).
(TIF)

**S7 Fig. Viral adhesion of MNoV-S7 particles to host cells.** (A) Viral initial attachment on the host cells (RAW264.7 or HEK293T/CD300lf) at 30 minutes after post-infection of the MNoV-S7 particles pretreated with DMEM or PBS(-)+EDTA (pH 8). After washing cells with the solution, the viral RNA was extracted from viruses attached on the cells and the amount was estimated by qRT-PCR. *($p < 0.05$). (B) Early genome replication of MNoV-S7 from 30 minutes to 12 hours. RAW264.7 cells infected with the MNoV-S7 particles pretreated with

DMEM or PBS(-)+EDTA (pH 8) were washed and collected at each time point of post-infection. RNA was extracted and the amount was estimated by qRT-PCR.
(TIF)

**S8 Fig. Multiple conformations of the MNoV-1 VLPs.** (A) A representative micrograph of the MNoV-1 VLPs. Scale bar, 500 Å. (B) MNoV-1 VLPs contain 3 types of particles; one large particle (T = 4), two small particles (T = 3) with the resting and rising P domain conformations. 3 types of particles were individually subjected to single particle analysis using RELION software. Table shows the number of selected parts after 2D classification, the number of particles used for final reconstruction, T-numbers, and resolutions in 3 types of the particles. (C) Representative 2D class average images in 3 types of particles. Scale bar, 200 Å. (D-F) 3D reconstructions of 3 types of the MNoV-1 VLPs; large particle (D), small particles with the resting (E) and rising (F) P domain conformations. Scale bar, 100 Å.
(TIF)

**S9 Fig. High-resolution cryo-EM map generation of the MNoV-S7 VLP suspended in DMEM, using a 300kV TEM.** (A) Representative micrograph of MNoV-S7 VLPs. Scale bar, 500 Å. (B) An example of 2D class average images. Scale bar, 200 Å. (C) A GS-FSC plot. Based on the 0.143 criterion for comparing two independent data sets, the resolution of the reconstruction is estimated to be 3.5 Å.
(TIF)

**S10 Fig. Structure comparison of the MNoV-S7 P domain by cryo-EM and the MNoV-1 P domain by X-ray crystallography.** For the crystallographic model, 3QLE (PDB ID) was used, which is the model of the closed state in A–B loop and E–F loop. Ribbon diagrams of the cryo-EM map of the MNoV-S7 P domain and the crystallographic model of the MNoV-1 P domain are colored magenta and gray, respectively, which are shown in the different angles. The cryo-EM model of the MNoV-S7 P domain was analyzed in the aqueous condition of the resting P domain conformation, while the crystallographic model of the MNoV-1 P domain was expectedly analyzed in the aqueous condition of the rising P domain condition (in main text). The large structural difference was found in the βC"–βD" loop (*) and the βC'–βD' loop (**) (S5 Fig), which make an interaction each other with those of the paired P domain in the P domain dimer. The cryo-EM model of the MNoV-S7 in DMEM shows the closed state in the βA"-βB" loop and βE"–βF" loop (***) (S5 Fig).
(TIF)

**S11 Fig. Elaborate interactions between the S domains of VP1 in the MNoV-S7 VLP.** (A) Amino acid sequence corresponding to the N-terminal of the S-domain. The residues indicated by the arrows (Gln19, Ala16, and Val30) are the start residues of the sequence for modeling the respective VP1's A, B and C monomers. Regions of the two short α-helices are indicated as αA and αB. In the lower panel, ribbon models of the S domain for the A, B and C monomers are superimposed. The residues colored blue, cyan, and purple correspond to the N-terminals of the A, B, and C monomers, respectively. (B) N-terminals of the A, B and C monomers on the S domain capsid are highlighted with the same colors in A. Asterisks indicate the pseudo-threefold axis of the T = 3 asymmetric units. (C) Interaction of the N-terminals of the S domains at the twofold axis. Asn49 and Asp52 of each C monomer form a polar interaction as shown by the dotted lines. (D and E) Interactions of the N-terminals at the pseudo-threefold axis in the T = 3 asymmetric unit and the fivefold axis, respectively. The counter residues at the adjacent N-terminals interact by hydrophobic bonds as indicated by the dotted lines. (F) Interaction of the N-terminal at the threefold axis. The N-terminal edge of the B monomer interacts with the adjacent loop in the C monomer by hydrophobic and polar

interactions as indicated by the dotted lines. (G) Comparison of NTAs of the S-domain in the A monomer among the members of the Caliciviridae family: MNoV (red), HNoV GI.1 (blue, PDB ID: 1IHM), RHDV (cyan, PDB ID: 3J1P), SMSV (yellow, PDB ID: 2GH8) and FCV (Green, PDB ID: 3M8L). Arrows indicate the direction of the extended NTAs. (H) the adjacent B and C monomers were added in (G) to show the elaborate crosslinks between monomers. The right panel is a schematic figure of the left panel. The NTA of S domain in MNoV (red), HNoV (blue) and RHDV (cyan) runs along the bottom edge of its S domain and interacts with the adjacent S domains, while the NTA in SMSV (yellow) and FCV (green) runs along the bottom edge of the adjacent S domain and interacts with the adjacent S domains.
(TIF)

**S12 Fig. Atomic model of MNoV-S7 in the rising P domain conformation.** (A) The atomic model modified and fitted to the 7.2 Å cryo-EM map of the MNoV-S7 infectious particle suspended in PBS(-)+EDTA (pH 8) is shown on the twofold axis. The left side only shows the map. (B) The map to model FSC curve (black line) showing a similar curve with GS-FSC (gray) indicates 8.5 Å resolution at 0.5 criteria.
(TIF)

**S13 Fig. Interactions between the rising P domain dimers in MNoV-S7.** (A) A cryo-EM map of the MNoV-S7 with the rising P domain conformation is overlapped with the modified atomic model (S12 Fig). The black triangle shows the threefold axis. S domains are colored black. The residues forming interactions between the P domains are colored yellow. C/C-dimers (purple) on the twofold axis form interactions with adjacent A/B dimers (blue) located around the fivefold axis. (B) The enlarged view of the dotted hexagon in A. (C) A stereo view of the candidate chemical interactions between the C/C and A/B dimers indicated by the dotted square in B. Pro425, Ser504, Leu524, and Gln526 of the C/C dimer are facing with Phe423, Gln263, Leu524, and Arg523 of the A/B dimer, respectively (S5 Fig), where hydrophobic and polar interactions are suggested to be formed.
(TIF)

**S14 Fig. Map and molecular model of the flexible hinge in the MNoV-S7 VLP having the resting P domain conformation.** (A) An enlarged view of the flexible hinge connecting the S and P domains. The flexible hinge was indicated by the dotted box. (B) The enlarged map and model view of the box in A. Arrow indicates a density between Glu223 and Asp231. (C) $Ca^{2+}$ as a possible metal ion is fitted into the density shown in B, and the distances from the near negatively charged residues are indicated.
(TIF)

**S15 Fig. A/B and C/C dimers structures of MNoV-S7 in the resting and rising P domain conformations.** (A and B) The ribbon model of the A/B dimer in the resting and rising P domain conformations, respectively. The S domain dimers show a bent conformation. The P domain dimer is tilted 10 degrees toward fivefold axis in the rising conformation. (C and D) The ribbon model of the C/C dimer in the resting and rising P domain conformations, respectively. The S domain dimers show a flat conformation. The resting P domain dimer changes to the rising conformation without a tilt.
(TIF)

**S16 Fig. Interaction of the P domain with the cellular receptor, CD300lf, in the two P domain conformations.** (A and B) The crystallographic model of the P domain and the CD300lf soluble domain (sCD300lf) (PDB ID: 5OR7) are fitted to the cryo-EM maps of MNoV-1 having the resting (A) and rising (B) P domain conformations, respectively. Three

sCD300lf molecules (magenta) are able to bind on the P domain inside the triangle (white dot lines) formed between one C/C dimer and two A/B dimers. In the resting P domain conformation, the triangle is distorted and the CD300lf molecules are more flexibly able to approach the binding site (A), while in the rising P domain conformation, the triangle is squeezed, limiting the space for the sCD300lf molecules to approach the binding site (B).
(TIF)

**S17 Fig. Structural stability of the HNoV GII.3 VLPs having the resting and rising P domain conformations.** (A) Representative micrograph of the HNoV GII.3 VLPs having the resting (blue circles) or rising (yellow circles) P domain conformations. (B) The HNoV GII.3 VLPs having the rising P domain conformation easily collapsed due to surface tension in thin vitreous ice, and only the stable VLPs having the resting P domain conformations were left. (C) In the ice embedded cryo-EM grid, the ice film is formed in a convex shape, where the center is thinner than the hole edge. Eventually, the unstable particles move to the edge or collapse.
(TIF)

**S1 Movie. Molecular mechanism of dynamic rotation of the P domain dimers.**
(MP4)

## Acknowledgments

We thank T.J. Smith for providing the 8 Å cryo-EM map of MNoV-1, T. Sato for helping the initial model building of the MNoV-S7, K. Namba and Y. Kawaoka for their helpful discussions, and G.C. Howard and R.N. Burton-Smith for critical reading of the manuscript.

## Author Contributions

**Conceptualization:** Kazuhiko Katayama, Kazuyoshi Murata.

**Data curation:** Chihong Song, Kazuyoshi Murata.

**Funding acquisition:** Kazuhiko Katayama, Kazuyoshi Murata.

**Investigation:** Chihong Song, Reiko Takai-Todaka, Motohiro Miki, Kei Haga, Akira Fujimoto, Ryoka Ishiyama, Kazuki Oikawa, Masaru Yokoyama, Naoyuki Miyazaki, Kenji Iwasaki, Kosuke Murakami, Kazuyoshi Murata.

**Supervision:** Kazuhiko Katayama, Kazuyoshi Murata.

**Visualization:** Chihong Song, Kazuyoshi Murata.

**Writing – original draft:** Chihong Song, Kazuyoshi Murata.

**Writing – review & editing:** Chihong Song, Kazuyoshi Murata.

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
