## [Decision Letter · Decision Letter 0]

20 Mar 2020

Dear Dr. Murata,

Thank you very much for submitting your manuscript "Dynamic rotation of the protruding domain enhances the infectivity of norovirus" for consideration at PLOS Pathogens. As with all papers reviewed by the journal, your manuscript was reviewed by members of the editorial board and by several independent reviewers. As you can see, the reviews are overall positive and the reviewers support your study. Nevertheless, the manuscript needs extensive revision to address the comments of the reviewers. This includes more detailed descriptions of experimental procedures and data analysis, but also proper discussion of recently published data to the same topic as indicated by one of the reviewers. In light of these comments, we would like to invite the resubmission of a significantly-revised version that takes into account the reviewers' comments.

We cannot make any decision about publication until we have seen the revised manuscript and your response to the reviewers' comments. Your revised manuscript is also likely to be sent to reviewers for further evaluation.

Sincerely,

Ralf Bartenschlager

Guest Editor

PLOS Pathogens

Guangxiang Luo

Section Editor

PLOS Pathogens

Kasturi Haldar

Editor-in-Chief

PLOS Pathogens

orcid.org/0000-0001-5065-158X

Michael Malim

Editor-in-Chief

PLOS Pathogens

orcid.org/0000-0002-7699-2064

Reviewer's Responses to Questions

**Part I - Summary**

Reviewer #1: The manuscript “Dynamic rotation of the protruding domain enhances the infectivity of norovirus” by Song et al. is generally well written and of high interest to the readers of Plos Pathogens. It provides a link between two previously noted conformations in noroviruses and clearly links the resting conformation to higher infectivity. It also clearly shows that metal binding is beneficial to infectivity. While of broad interest, the manuscript requires a major revision including discussion of results by some other groups prior to publication.

Reviewer #2: Here Song, Takai-Tokada et al provide 8 cryo-EM reconstructions (7 at middle to low resolution) of norovirus capsids : 4 of infectious murine norovirus for two strains (MNV-1 and MNV-S7) in 2 conformations each, 2 of a human norovirus (genotype II.3) VLP again in 2 conformations, 1 of MNV-S7 VLP in complex with the soluble part of the murine norovovirus receptor CD300lf, and finally a high resolution (3.5 A) VLP of MNV-S7. There are now numerous structures of norovirus capsids and several have been solved to high enough resolution that atomic models could be built: One X-ray structure for a human GI.1 VLP (Prasad et al., 1999) and several cryo-EM structures in recent years (e.g. the GI.1, GI.7, GII.2 and GII.4 atomic structures in Jung et al., PNAS 2019). Previously, depending on the strain, the spike domains were found in two distinct conformations (here called by Song et al. 'rising' (Ri) and 'resting' (Re)). It has long been suspected in the field that all norovirus capsids can actually assume both conformations (e.g. indirect evidence from the T.J. Smith and G. Hansman labs). Here Song et al. establish this fact for MNV-1 and MNV-S7 particles and for GII.3 VLP (though the lower resolution reconstructions for the latter would still leave some doubt, if not for the rest of the evidence). This, and the triggers used by Song et al., are major findings and should be better emphasized in the paper. In general, the paper should be carefully rewritten following the suggestions below.

Reviewer #3: This paper by Murata, Katayama, and coworkers describes Cryo-EM based analysis of MNV capsids at different pH values and in the presence and absence of bivalent metal ions. The experimental data appear to be sound and the presentation of the results is adequate. The study has led to interesting new results that further our understanding of the MNV infection process. Essentially, the authors demonstrate that the „rising“ and „resting“ conformations of the P-domain can be switched by altering pH and the concentration of bivalent metal ions. They hypothesize that this conformational mimicry is linked to disassembly or assembly of virus capsids as an essential step during the infection process. Rising and resting conformations of Calicivirus capsids, including Norovirus, have been described before, but this is the first time that interconversion between the conformational states has been observed as being a direct result of a changing chemical environment. Importantly, the study links these structural features to binding and infection assays. As a major result, the „resting“ conformation apparently binds better to target cells and leads to more efficient infection. The topics addressed certainly match the audience of PLOS Pathogens very well. Therefore, I definitely support publication.

**Part II – Major Issues: Key Experiments Required for Acceptance**

Reviewer #1: Major:

- On July 04 2019 Snowden,..., Stonehouse published a study on biorxiv (https://www.biorxiv.org/content/10.1101/693143v1), 5 months before this study appeared on biorxiv. Here, also elevated P domains were observed in MNV, which rotated compared to the resting state. The study used heat treatment instead of pH. The results should be linked to and discussed with those presented here. In light of these results, l. 73/74 require rephrasing.

- Snowden et al reported a 3.1 A resolution structure. For the rising P domain, the P dimers had to be modelled separate from the shell. Little detail is provided by the authors, how they got to their 5+ A resolution structures. More detail on the reconstruction process has to be provided in the methods section. Moreover, was part of the dataset retained to check for overfitting of the data.

- HNV is not generally used for human noroviruses, hNoV or huNoV should rather be used.

- In 2019, Mallagaray et al. reported a deamidation in some huNoV-like particles. This is strongly linked to pH. Have the authors any idea, whether MNV gets deamidated? Partial deamidation could alter kinetics and explain, why particles convert so slowly back to the resting conformation. Additionally, huNoV particles had been shown to often be instable at alkaline pH (e.g. Pogan et al. 2018). This was shown to be influenced by slight sequence variations as observed here for the two MNV strains (p. 10). This should be mentioned.

- Jung et al. 2019 also observed rising and resting conformations in huNoV-like particles. Even more relevant to the here presented study Koromyslova et al. observed in 2015 structural alterations in presence of citrate (Treatment of norovirus particles with citrate), which resemble a lot the micrographs of the rising domain in Fig. S1. Do the authors consider these structures similar? It would present alternative strategies to achieve the rising conformation in huNoVs, which are not known to bind cations. What is the fraction of broken/disassembled particles observed?

- The authors only touch briefly on VP2 and genome. It should be stated more clearly if in any of the conformations of MNV-1 or MNV-S7 virions extra density was observed in asymmetrical reconstructions that could be attributed to VP2 or genome-protein contacts.

- DMEM is rather complex compared to PBS. It would be important to also show particles in PBS +/- EDTA at pH 8 and also at lower pH to more clearly pinpoint the effect to pH.

- The structural conversion is super slow with 2-8 h and also the mixed species that are observed are super-interesting. Have the authors a theory for the slow kinetics? Could the P domain connections stabilize a given conformation causing hysteresis? While these aspects are discussed, they warrant a more thorough reflection in terms of potential mechanisms, timeline of infection (binding, entry and genome release). I hence recommend extending the discussion of these aspects.

- What was the reasoning to use such closely related MNV strains? A more distant MNV would have been of higher interest to establish a general relationship between rising/resting conformation and infection.

- 40-50 nm VLP of MNV-1 suggests assembly defects or formation of T=4 particles. Those have been observed in huNoVs previously, e.g. in GII.4 strains by Devant et al 2019. While the larger particles can be the result of suboptimal production conditions, it would be of interest to show those micrographs and explain what the larger particles are.

- Upon lifting and rotation, are there any structural changes within the P domain backbone? This should be mentioned explicitly.

- Have the authors an idea why CD300lf binds so poorly? Have they tested whether bile acids or glycans enhance the binding of the protein receptor? This could be discussed in a bit more detail. How would the reconstruction look without imposing symmetry? Maybe the receptor would be less blurred out?

- Fig. S9C: electron density seems to indicate a subunit interaction that is not well reflected in the model, i.e. Ser-Gln maybe positioned lower or are in fact other residues interacting? The two lower interactions lack density entirely. How certain are these?

- How different are the positions of the rising P domain from each other in the three different strains/particles? An overlay would be useful.

As side note, with the wealth of supplementary data and the caption being given after the references, it was difficult to review the manuscript. I would have appreciated two files and captions closer to the figures.

Reviewer #2: (No Response)

Reviewer #3: None

**Part III – Minor Issues: Editorial and Data Presentation Modifications**

Reviewer #1: Minor:

- last sentence of abstract is difficult to understand and requires rephrasing

- l. 221 vs. l. 227 appear contradictory. The fine structure discussion is not fully clear from the phrasing.

- l. 500 A/B dimer not diner

- Fig. S1 caption says PBS(-)-EDTA whereas figure says +EDTA. The latter makes more sense to the referee. The authors should check whether this has been correctly stated throughout text.

- Fig. S6: comparing MNV sequences to huNoV GII.3 would be more useful to estimate how different these are. What is the rationale for showing the Norwalk sequence?

- Fig. S10: please also label the other residue present in B and C. Is it an Arg and how far is its functional group away from the putative Ca2+? There is an additional residue, presumably from the helix, overlaying with Asp 231. If this is not close to the metal binding, consider taking this out to increase clarity of the view.

Reviewer #2: 1) I suggest merging figures 1 and 5 so the reader grasps at once that the two conformations are found in both MNV-1 (for which only Ri was reported so far) and GII.3 VLPs.

2) Count of particles in Ri and Re conformations: This is another important piece of evidence. Both conformations are found on the same grids and depending on buffer conditions their relative abundance changes (appendix figures S1 I, S4 K). Furthermore a time course of going from Ri to Re is actually given for MNV-1, with indication that mixed particles (part Ri, part Re) can be detected (S3 Fig).

a) Most important: How are the particles counted ? I could not find that information in the paper. I am guessing from e.g. S3 Fig. that during image processing 2D classes are used to assign particles to Ri, Re or mixed. Or was this done by eye on random subsets of extracted particles ? If so, which set of particles was used to draw the subsets ? Please give a clear and detailed description of the process.

b) On a related note, very few particles per image are actually used in the 3D reconstructions. For instance

l. 448 'For MNV-1 infectious particles suspended in DMEM and PBS(-)-EDTA (pH 8.0), 6,708 and 4,704 particles were selected from 1,046 and 2,188 images, respectively'.

Yet e.g. the inserts from micrographs in S1 Fig. show 6-12 seemingly good particles, so most are rejected during image processing. Please give numbers: How many 'not junk' particles (seemingly good particles) don't make it to the final 3D reconstruction ? Discuss in terms of heterogeneity (mixed spike conformations, departure from icosahedral symmetry, broken particles, ...).

c) Make a separate, main figure from appendix figures S1 I, S4 K and S3. This could be new Fig. 2.

3) l. 345 'HNV GII.3 VLPs also showed that the rising P domain conformation easily collapsed due to surface tension of thin vitreous ice, thus leaving most of the particles with a resting P domain conformation in the thin ice of cryo-EM (S14 Appendix).'

l. 896 'The VLPs having the rising P domain conformation easily collapsed due to the surface tension in thin vitreous ice, and only the stable VLPs with the resting P domain conformations were left.'

I do not understand the basis of these statements. Do the authors suggest that the spikes collapsed between sample deposition and plunge freezing ? Why ?

4) l. 291-314 in the discussion section actually belong in the results.

5) l. 276 'metal ions binding to the region between the P and S domains, may help to stabilize the resting

277 conformation (S10 Appendix).'

The evidence for this is that EDTA-containing buffers seem to favor Ri, while DMEM favors Re. However, pH seems to be the most important factor.

a) What is the metal ion content of DMEM ?

b) It could be worth mentioning that the GI.1 crystal structure, until very recently the only Re conformation, was obtained at pH4.8.

c) l. 279 'we identified density between the two carboxylates (Glu223 and Asp231) in the flexible hinge of the resting P domain conformation, where a metal ion is expected to bind to these residues (S10 Appendix)'. The density shown in S10 Fig is not convincing. At this threshold the map shows other unexplained features, suggesting that it is contoured too low. Please remove this statement.

6) The CD300lf data are not convincing.

a) l. 875 'rCD300lf is fitted to the extra density using the docking simulation with a parameter set shown in Materials and Methods'. If I understand correctly, the authors perform ab initio docking, find a pose that roughly matches the extra density, and conclude that they have the approximate position of the receptor binding site. However, the extra density is clearly not that of rCD300lf and the docking pose does not match the crystal structure of the complex of the receptor with the spike (Kilic et al., J. Virol. 2018). This part should be discarded entirely.

b) Discussion of the accessibility to rCD300lf in Ri vs Re is relevant here but should simply match the available crystal structure to the Re and Ri reconstructions.

c) l. 329 'The resting P domain conformation showed higher accessibility of the cellular receptor CD300lf than the rising P domain conformation, causing greater viral infectivity.' As I remember Kilic et al. did not find any hindrance to binding to the MNV (Ri) conformation. But even then the 'causing greater viral infectivity' would be speculative.

7) l. 218 'The flexible N-terminals formed various interactions with each other between adjacent S domains at the twofold, threefold, pseudo-threefold, and fivefold axes (S8 Appendix B-F)'

This is a secondary point in the paper, but S8 Fig. is a very nice recap of the N-terminal arm's differing positions in Caliciviridae (same subunit vs domain-swapped). The authors should point out that the 5-fold (molecule A) vs 3-fold (i.e. quasi-6-fold, molecule B) positions of the arms are determinants of quasi-equivalence between pentamers (5 'A' arms) and hexamers (3 'B' arms), with the 'C' arms always more disordered.

8) Minor corrections/clarifications

a) l. 148 'homology' shoud be 'sequence identity'

b) l. 390-392 'two days after infection. MNV was precipitated using ultracentrifugation with a 30% sucrose

cushion'

'diluted with DMEM without any additional supplements and precipitated again to remove CsCl.'

Replace 'precipitated' with 'pelleted'.

c) S2 Table define 'Ri' 'Re'

d) Fig. 1 '(red arrows in A and C)' ... '(red arrows in B and D)' are almost invisible at printed size (especially A and C)

e) S2 Fig. '(B) The cells' please indicate 'HEK293T/CD300lf cells'. In general label the figures so that it is immediately clear which of the two cell types was used.

f) l. 429 '(C) The early genome replication in each cell incubated for 0, 2, 4, 6, 8, 10, and 12

hours post-infection.' Please describe the protocol. I could not find it in the methods. Presumably cells were pelleted and lysed to separate intracellular from extracellular RNA?

g) l. 771 'MNV-S7 pretreated with PBS(-)-EDTA (pH 8) produced slower propagation of the virus than

those pretreated with DMEM, though the difference was small compared to that of MNV-1 (Fig

1E).' All points are actually within error bars. The PBS(-)-EDTA (pH 8) curve is systematically below though, so an appropriate test (Mann-Whitney rank test ?) may be helpful here.

h) l. 815 'Asn49 and Asp52 of each C monomer form a charged interaction as shown by the dotted lines'. None of the interactions shown are actually charge-charge (at most one charged residue involved). Please change all 'charged' to 'polar' or 'likely hydrogen bond'.

Reviewer #3: There are points that should be addressed by the authors before publication. It has been described by others that Ca2+ improves the binding affinity of MNV P-domains for CD300lf by about one order of magnitude, with bile acid, i.e. GCDCA adding another factor of two (Nelson et al., PNAS 2018). The respective crystal structure has well defined binding sites for Ca2+. The authors of that paper discuss the possibility that binding fo MNV to CD300lf is linked to conformational changes of the A’-B’ and the E’-F’ loops, which exist in an open and closed conformation. From this, two questions arise immediately. First, have the present experiments been performed in the absence or in the presence of GCDCA? Second, could the improvement of cell attachment of the resting conformation be the consequence of a combined effect, i.e. „loop conformation“ plus „overall conformation“?

There is no information about the concentrations of bivalent metal ions, except that binding behavior changes upon addition of EDTA. It would be very helpful to have that information somewhere in the Materials & Methods section.

PLOS authors have the option to publish the peer review history of their article (what does this mean?). If published, this will include your full peer review and any attached files.

Reviewer #1: No

Reviewer #2: No

Reviewer #3: Yes: Thomas Peters
---

## [Decision Letter · Decision Letter 1]

11 May 2020

Dear Dr. Murata,

After internal review of your revised manuscript and re-review by reviewer #1, we are pleased to inform you that your manuscript 'Dynamic rotation of the protruding domain enhances the infectivity of norovirus' has been provisionally accepted for publication in PLOS Pathogens.

Before your manuscript can be formally accepted you will need to complete some formatting changes, which you will receive in a follow up email. A member of our team will be in touch with a set of requests. In addition, during this process, please take again a close look at your manuscript to remove the typos that are still contained in your manuscript.

Best regards,

Ralf Bartenschlager

Guest Editor

PLOS Pathogens

Guangxiang Luo

Section Editor

PLOS Pathogens

Kasturi Haldar

Editor-in-Chief

PLOS Pathogens

orcid.org/0000-0001-5065-158X

Michael Malim

Editor-in-Chief

PLOS Pathogens

orcid.org/0000-0002-7699-2064

Reviewer Comments (if any, and for reference):

Reviewer's Responses to Questions

**Part I - Summary**

Reviewer #1: The authors have satisfactorily addressed my comments in their revision, I hence recommend publication in PLOS Pathogens.

As a side note, the revision still contains some typos.

**Part II – Major Issues: Key Experiments Required for Acceptance**

Reviewer #1: (No Response)

**Part III – Minor Issues: Editorial and Data Presentation Modifications**

Reviewer #1: (No Response)

PLOS authors have the option to publish the peer review history of their article (what does this mean?). If published, this will include your full peer review and any attached files.

Reviewer #1: No